# Anatomical Features of the Parotid Duct in Sialography as an Aid to Endoscopy—A Retrospective Study

**DOI:** 10.3390/diagnostics12081868

**Published:** 2022-08-02

**Authors:** Gal Avishai, Muhammad Younes, Hanna Gilat, Leon Gillman, Vadim Reiser, Eli Rosenfeld, Gavriel Chaushu, Daya Masri

**Affiliations:** 1Rabin Medical Center, Department of Oral and Maxillofacial Surgery, Beilinson Hospital, Petach Tikva 49414, Israel; leongi@clalit.org.il (L.G.); vadimre@clalit.org.il (V.R.); elir2@clalit.org.il (E.R.); gavrielce@clalit.org.il (G.C.); dayama@clalit.org.il (D.M.); 2Department of Oral and Maxillofacial Surgery, The Maurice and Gabriela Goldschleger School of Dental Medicine, Tel-Aviv University, Tel Aviv 69978, Israel; muhammady@mail.tau.ac.il; 3Rabin Medical Center, Department of Otolaryngology-Head and Neck Surgery, Beilinson Hospital, Petach Tikva 49414, Israel; hannagi2@clalit.org.il

**Keywords:** salivary gland endoscopy, sialography, parotid duct anatomy

## Abstract

Sialography is used for diagnosis of obstructive salivary gland diseases and prior to sialendoscopy. Three-dimensional cone beam computerized tomography (CBCT) sialography allows imaging and measurement of salivary duct structures. Salivary gland endoscopy has a long learning curve. The aim of this retrospective study is to create an anatomical quantitative guide of different distances and angles significant for endoscopy. Twenty-six CBCT sialographies of healthy parotid ducts were included. Outcome parameters included diameters, distances, angles and number of minor tributaries. Results show the average distance from the papilla to the curvature of the gland was 41.5 mm (Q1 36.97 mm–Q3 45.32 mm), with an angle of 126.9° (Q1 107.58°–Q3 135.6°) of the curvature and a distance of 35.25 mm (±7.81 mm) between the curvature and the hilus. The mean width of the duct was 0.8 mm (Q1 0.7 mm–Q3 1.15 mm) at its narrowest and 2 mm (Q1 1.4 mm–Q3 2.2 mm) at its widest. This is the first anatomical quantitative study of the parotid duct in relation to sialendoscopy.

## 1. Introduction

Obstructive diseases of the salivary glands are their most common pathology. Its incidence is 1:10,000 to 1:30,000, comprising 30 percent of salivary pathologies [1,2].

The diagnosis and treatment are based on imaging modalities such as ultrasound (US), plain radiography and sialography, magnetic resonance imaging (MRI), computed tomography (CT) and nuclear scintigraphy/positron emission tomography (PET) [3].

The present study is based on sialography of the parotid gland duct (Stensen’s duct). Sialography was initially described in 1902 by Carpy and remains the most accurate method for imaging the salivary ductal system. It provides a cost effective and safe diagnostic solution that can be routinely employed as a screening procedure for salivary gland pathologies, namely aberrations in the ductal system, sialoliths and Sjögren’s syndrome [4,5].

The standard used today is sialography under cone-beam CT (CBCT), following injection of contrast agent into the ductal system. Once the fat-soluble radiopaque contrast agent has filled the duct system, the CBCT scan allows the demonstration of morphology and pathology of the main ducts, the minor tributaries and the parenchyma of the gland. CBCT is being used increasingly for point-of-service head and neck and dentomaxillofacial imaging [6,7]. There are limited contraindications for sialography, the main one being known allergy to contrast material. In case of acute sialadenitis, sialogrpahy is also contraindicated, since infusion of the contrast material may worsen the infection [8].

Since the 1990s, semi-rigid miniature endoscopes have been used for diagnoses and treatment of salivary duct pathologies, and endoscopy of the salivary gland has become the standard of care [9]. The first stage of the endoscopy is locating the punctum or papilla of the parotid gland duct (parotid duct). The punctum is serially dilated with conical dilators and probes of increasing size. After adequate dilatation, the sialendoscope is introduced through the dilated punctum. In order for the endoscope to demonstrate a clear image of the lumen of the duct, it has to be dilated by constant irrigation with pressure, achieving an optical cavity. The constant irrigation also washes out the debris and sludge accumulated in the ductal system, thus increasing the intraductal visibility [2]. The operator must follow the route of the duct while maintaining the center of the lumen of the duct in the center of view of the endoscope. A few maneuvers are needed while progressing through the duct, namely a pivotal movement when the duct curves around the masseter muscle and another pivotal movement in order to enter the tributaries branching from the main duct. A number of miniature instruments are available for use with sialendoscopes including grasping forceps, biopsy forceps, micro drills, wire baskets, high pressure balloon dilators and guide wires [10].

To date, surgical anatomy of the parotid duct has been studied from the point of view of open parotid surgery, mainly in relation to the branches of the facial nerve [11,12], in two-dimensional sialography [13] and for the assessment of parotid tumors [14].

The aim of the present study was to assess quantitative radiographic anatomy of the parotid duct based on three-dimensional CBCT sialography in order to create quantitative guidelines that will aid in endoscopy of the parotid duct.

## 2. Materials and Methods

### 2.1. Study Group

This study is a retrospective analysis of all sialographic examinations performed under cone beam CT at the outpatient clinic of the department of Oral and Maxillofacial Surgery at Rabin Medical Center—Beilinson Hospital, between the years 2018 and 2022. The study was conducted according to the guidelines of the Declaration of Helsinki and approved by the Institutional Review Board of Rabin Medical Center (protocol code RMC-22-0218, date of approval 28 March 2022).

### 2.2. Sialography Technique

Sialographies were performed in a standard manner; a 26 G lacrimal cannula (BVI Visitec^®^, CityPoint Waltham, MA, USA) was introduced into the parotid duct, and 3 cc of X-ray contrast medium (Omnipaque^TM^ 350 mg/mL, GE Healthcare, Cork, Ireland) was infused into the duct. A cone beam CT was acquired under constant digital pressure of the contrast medium. All sialographies were performed by the same physician (G.A.).

### 2.3. Inclusion Criteria

Patients over 18 years old.Available sialography demonstrating the entire length of the parotid gland duct.

### 2.4. Exclusion Criteria

Insufficient demonstration of the parotid gland duct.Mega duct—a pathological status of the duct, which renders length measurement impossible (Figure 1).Distinct sialectases (widening of the proximal ducts).Known diagnosis or sialographic features of Sjögren’s syndrome.Rupture of the duct during infusion of the contrast medium.

### 2.5. Data Collection

Measurements were performed using the PACS system of Rabin Medical Center (Vue Pacs Version 12.1.5, Carestream Health Inc.©, Rochester, NY, USA), which allows for accurate measurements of lengths and angles. All study images were positioned in a standard fashion with the head located at the true lateral position. Figure 2 shows an example of a normal parotid duct obtained from CBCT sialography.

Measurements taken were chosen in accordance with relevant steps of endoscopy: 1st step—from papilla to the curvature of the duct, in which the progress is in a straight route, the distance from the papilla to the center point of the curvature was measured; 2nd step—pivot of endoscope at the curvature, the angle of curvature was measured; 3rd step—advancement from curvature to hilus, which is also in a straight route, distance between the center point of curvature to the furcation of duct (hilus) was measured; 4th step—exploration of each of the main tributaries at the hilus, the angle of furcation between each of the two main tributaries and the main duct was measured. The duct’s diameter was measured to assist in choosing endoscope size. The number of minor tributaries which confluence with the duct distally to the hilus was counted; these minor ducts are usually not amenable to endoscopy since they are too narrow or at an acute angle to the main duct.

Linear data were acquired by the 3D line measure tool of the PACS system (Figure 3). Angle measurements (Figure 4 and Figure 5) for assessing the flexure of the duct and its tributaries were performed using the angle tool. Four angles were defined: α—the angle of the curvature, β—angle between the parotid duct (PD) and superior parotid duct (SPD), γ—angle between parotid duct and inferior parotid duct (IPD), δ—angle between superior and inferior parotid ducts.

Data were collected in a standardized Excel file, and all measurements were performed by one examiner (M.Y.). Data were grouped by age and gender.

### 2.6. Data Analysis

Data were analyzed using IBM SPSS© Statistics for Windows, version 25.0 (IBM Ltd., Armonk, NY, USA). For each measurement an analysis of normal distribution was performed. When the distribution was normal average and standard deviation were used. When the distribution was not normal an interquartile range was calculated and the mean defined as 50%.

## 3. Results

The study group consisted of 113 cone beam CT-sialographies. After applying the exclusion criteria, 26 sialographies were included. The age range was normally distributed; the mean age of the patients was 53.92 years with a standard deviation of 16.96 years, the youngest was 21 years old and the oldest 80. Twenty of the patients included were female (76.92%), and 6 were male (23.08%).

The features of each duct were measured as described in the Methods section and are presented in Table 1. Where normal distribution of the data was found, the average and standard deviation were used, and when the data were not normally distributed the interquartile range is presented. No significant differences were found between males and females or with correlation to age.

## 4. Discussion

This study aimed to describe the anatomy of the parotid duct as an aid to endoscopic treatment using CBCT sialographies of healthy parotid glands. To the best of our knowledge this is the first description of quantitative duct anatomy. Endoscopy of the parotid duct requires a long learning curve, and anatomical landmarks using average duct anatomy may aid in the learning process.

Anatomy of the parotid duct has been studied mostly by cadaver dissection with the focus of interest being the relation of the duct to the branches of the facial nerve within the parotid gland. One cadaver study found that the buccal branch of the facial nerve was inferior to the parotid duct in 75% of 20 glands [11]. Another cadaver study focused on the anatomy of the duct and found that 31% of the glands had a single duct from the papilla to within the gland and that 62% of the ducts had a branching system, and of those 48.3% were bifurcated and the rest were trifurcated or had a branching system [12]. In our study group all the ducts had multiple furcations, and not a single one had a single duct; this difference exemplifies the difference in the study methodology, where in cadaver studies minor tributaries can be missed within the parenchyma of the gland.

In a study of healthy morphometry of the salivary ducts using digital subtraction sialography of 43 parotid ducts under fluoroscopy [13], the length and width of the parotid duct were measured. The mean width of the duct was found to be 1.6 mm, similar to our results. The average length of the duct was 50 ± 9.6 mm (although the article does not state the start and end point for this measurement), and this is markedly different to our finding of a 75.4 mm mean between papilla and hilus. This difference may be attributed to a different definition of the end point of the duct.

As a guide to endoscopic approach of the parotid gland duct, using the mean values found, an average trajectory of advancement of the endoscope within the duct is as follows (Figure 6): In the first step after entering the duct through the punctum, move forward for an estimated distance of 41.5 mm (Q1 36.97 mm–Q3 45.32 mm) before reaching the curvature. In the second step, a pivot movement of the endoscope laterally and superiorly towards the gland is needed to surpass the curvature of the duct around the masseter muscle, and this is especially important since forceful manipulation in this area may cause duct perforation. The angle of the curvature is an estimated 126.9° (Q1 107.58°–Q3 135.6°). In the third step, progression from the curvature to the hilus is linear, and the distance between them is an estimated 35.25 mm (±7.81 mm). During progression in the third step, entering accessory minor ducts/tributaries will most likely not be possible because of their acute angle to the main duct. At the hilus of the gland where the two main tributaries split, move superiorly to explore the superior parotid duct at an angle of 123.68° (±22.68°) and inferiorly towards the inferior parotid duct at an angle of 136.94° (±25.22°). The endoscope chosen should be at a diameter of around 1 mm to pass through most of the ducts and should still be sturdy enough to allow advancement through narrower ducts.

This anatomical study of the parotid duct aimed to depict the normal anatomy based on 3D sialography. One limitation of the present study is the relatively small sample size which can be explained by the fact that most sialographies performed are for pathological ducts and glands. In our group only 23% of the ducts were found to be normal and were included in the study.

Future directions for this line of research can include an increase in the study group with a focus on differences between males and females, changes with age and the effect of specific pathologies on the anatomy of the duct of the parotid gland.

## 5. Conclusions

Cone beam sialography of the parotid duct aids in the endoscopic approach and the learning curve of the technique. Average measurements of the distances and angles of the duct are presented in this study.

## Figures and Tables

**Figure 1 diagnostics-12-01868-f001:**
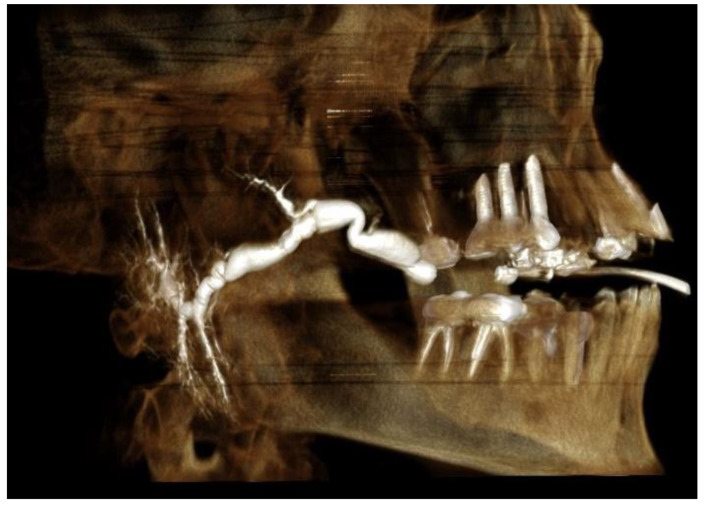
CBCT sialography of right parotid gland exhibiting mega duct.

**Figure 2 diagnostics-12-01868-f002:**
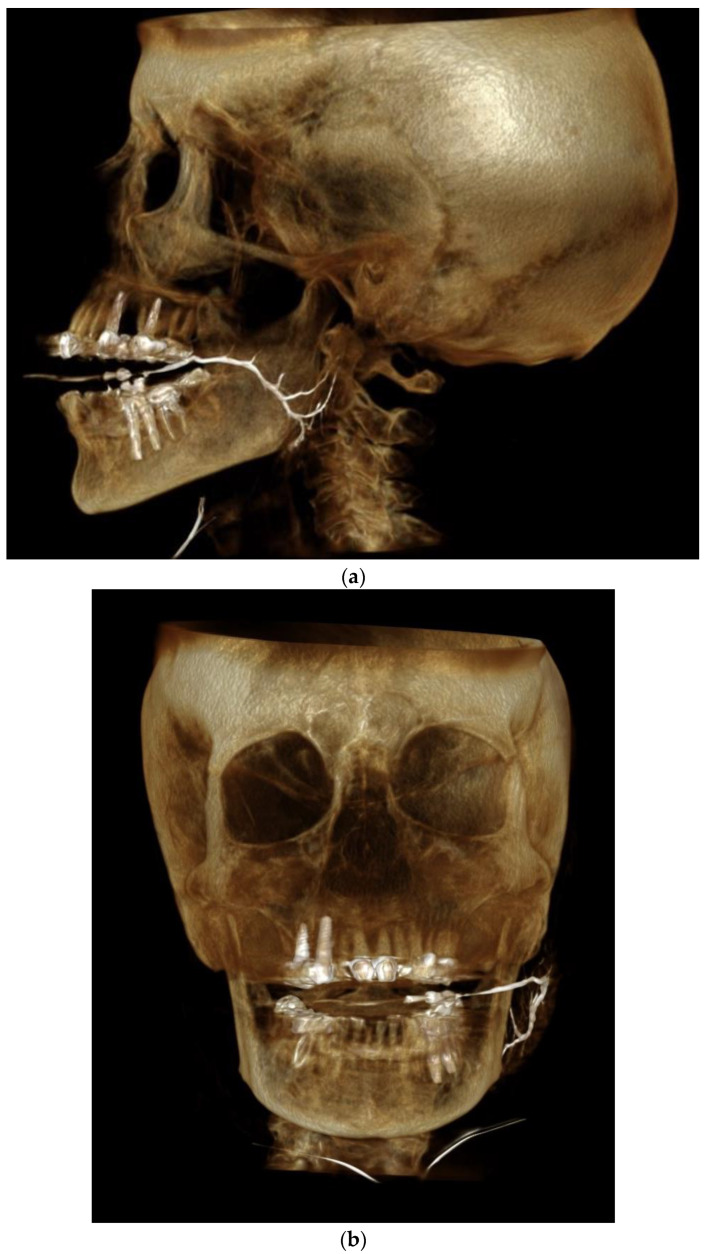
(**a**). Healthy parotid duct on cone beam CT sialography—lateral view. (**b**). Healthy parotid duct on cone beam CT sialography—anterior view.

**Figure 3 diagnostics-12-01868-f003:**
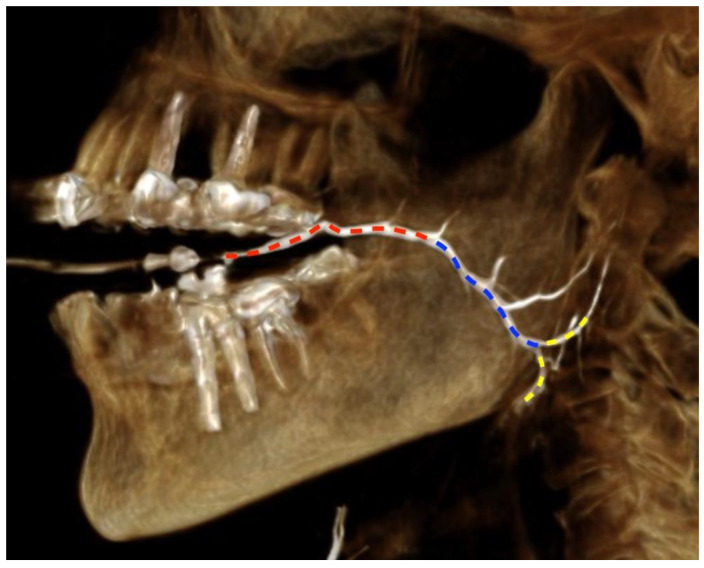
Parotid duct distances measurements. red: distance between papilla and curvature, blue: distance between curvature and hilus, yellow: bifurcation after duct hilus.

**Figure 4 diagnostics-12-01868-f004:**
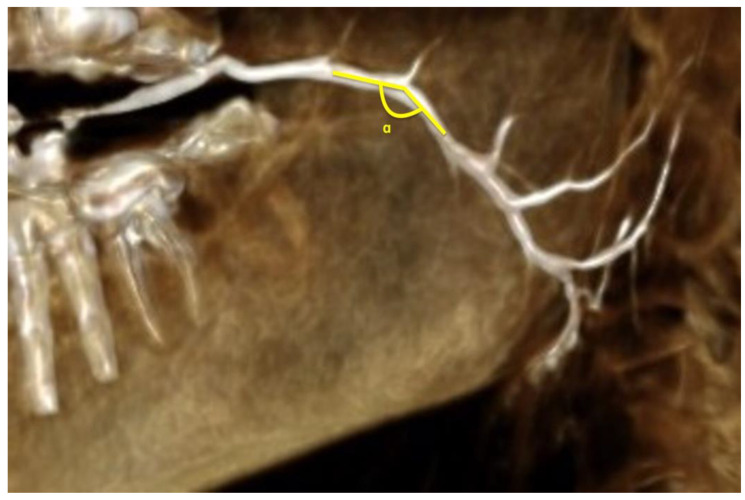
Angle of curvature of parotid duct (α).

**Figure 5 diagnostics-12-01868-f005:**
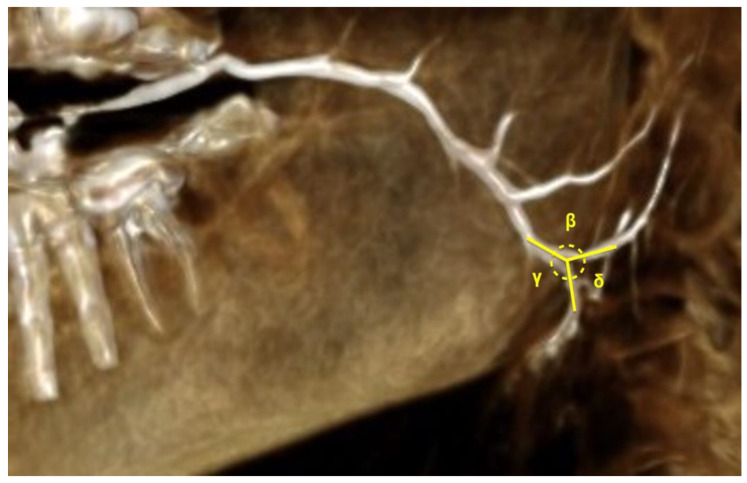
Angles between parotid duct and its tributaries after hilus. (β) Angle between parotid duct and superior parotid duct; (γ) angle between parotid duct and inferior parotid duct; (δ) angle between superior and inferior parotid duct.

**Figure 6 diagnostics-12-01868-f006:**
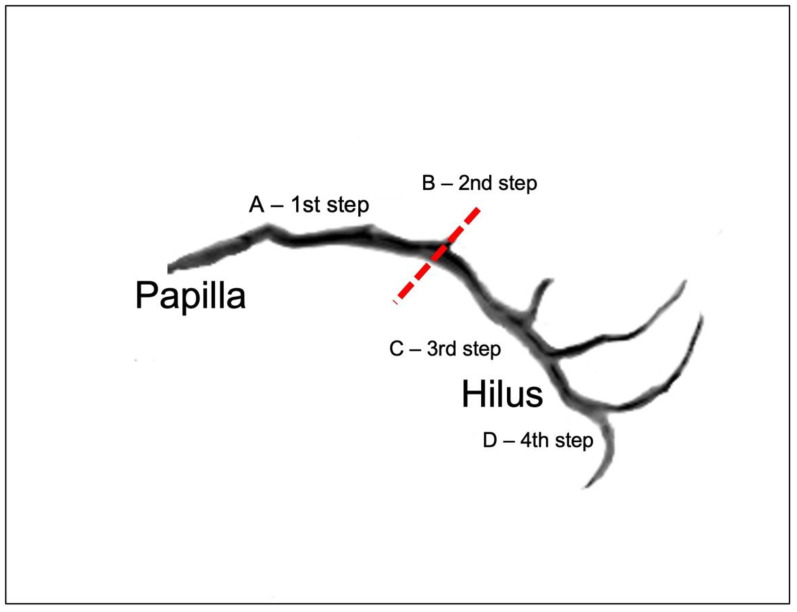
Steps of parotid duct endoscopy in relation to duct anatomy. Dotted red line—mid-point of curvature.

**Table 1 diagnostics-12-01868-t001:** Features of the parotid duct.

	Average	SD	Minimum	Maximum	Normal Distribution	Q1 (25%)	Q2 Median (50%)	Q3 (75%)
**Distance Between Papilla and Curvature**			35.4	52.9	NO	36.97	41.5	45.32
**Angle of Curvature (** α **)**			69.3	155	NO	107.58	126.9	135.6
**Distance Between Curvature and Hilus**	35.25	7.81	19.4	53.8	YES			
**Distance Between Papilla and Hilus**			63.6	100.6	NO	71.2	75.4	80.77
**Number of Tributaries Pre-Hilus**			0	4	NO	0.75	2	2
**Number of Tributaries Post-Hilus**			2	3	NO	2	2	2
**Angle between PD and SPD (β)**	123.68	22.68	83.11	167.9	YES			
**Angle between PD and IPD (γ)**	136.94	25.22	64.54	177.48	YES	124.53	140.88	154.62
**Angle between SPD and IPD (δ)**			41.5	202	NO	60.59	90.6	130.99
**Duct Width—Maximal**			0.9	4	NO	1.4	2	2.2
**Duct Width—Minimal**			0.5	2.5	NO	0.7	0.8	1.15

SD—standard deviation, PD—parotid duct, SPD—superior parotid duct, IPD—inferior parotid duct.

## Data Availability

The data presented in this study are available on request from the corresponding author.

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
