# Peer review of "Anatomical Features of the Parotid Duct in Sialography as an Aid to Endoscopy—A Retrospective Study"

_diagnostics, 2022, doi:10.3390/diagnostics12081868_

Round 1

Reviewer 1 Report

The paper presents a sound study of high relevance. You did not include the term retrospective which should be in the title and the text.  A power analysis to justify the statistics might be difficult to perform and 26 CBCT scans are a fair amount. Wonder if the literature covering anatomical studies could give you some information for the follow-up study conc power estimation and anatomical measures. You have supplied the paper with nice illustrations, however an AP illustration would complete the 3 D impression. Could you supply the paper with this info?

Author Response

Dear Reviewer,

Thank you so much for your positive review and insightful suggestions.

Please see the following responses to your queries:

  • The paper presents a sound study of high relevance.
    • Thank you
  • You did not include the term retrospective which should be in the title and the text.
    • Added to title, abstract and text (M&Ms)
  • A power analysis to justify the statistics might be difficult to perform and 26 CBCT scans are a fair amount. Wonder if the literature covering anatomical studies could give you some information for the follow-up study conc power estimation and anatomical measures.
    • Thank you for this suggestion but we did not find another published article measuring the features of the parotid duct measured here. As stated in our discussion, the closest studt was Horsburgh, A.; Massoud, T.F. The salivary ducts of Wharton and Stenson: analysis of normal variant sialographic morphometry and a historical review, but they only measured the total length.
  • You have supplied the paper with nice illustrations, however an AP illustration would complete the 3 D impression. Could you supply the paper with this info?
    • Added

We are happy to answer any other concerns you might have

Reviewer 2 Report

This is an excellent paper presenting well-designed research. The manuscript is concise, graphic material is very good and helpful in understanding the measurements taken. Table 1 has great practical merit. Discussion proves the originality of this study. As salivary gland endoscopy is getting more popular, this research might set golden standards for those performing sialendoscopy.  I have only one remark, concerning references; 8 out of 12 articles are more than 10 years old. I would recommend finding some more recent papers.     

Author Response

Dear Reviewer,

Thank you so much for your positive review and insightful suggestions.

Please see the following responses to your queries:

  • This is an excellent paper presenting well-designed research. The manuscript is concise, graphic material is very good and helpful in understanding the measurements taken. Table 1 has great practical merit. Discussion proves the originality of this study. As salivary gland endoscopy is getting more popular, this research might set golden standards for those performing sialendoscopy. 
    • Thank you !
  • I have only one remark, concerning references; 8 out of 12 articles are more than 10 years old. I would recommend finding some more recent papers. 
    • Agree with this remark. Added refs 5. and 7. Alas, most of the recently published studies deal with sialendoscopy and very few with sialography.  

We are happy to answer any other concerns you might have